# Performance Metrics of the Scoring System for the Diagnosis of the Beckwith–Wiedemann Spectrum (BWSp) and Its Correlation with Cancer Development

**DOI:** 10.3390/cancers15030773

**Published:** 2023-01-26

**Authors:** Maria Luca, Diana Carli, Simona Cardaropoli, Donatella Milani, Guido Cocchi, Chiara Leoni, Marina Macchiaiolo, Andrea Bartuli, Luigi Tarani, Daniela Melis, Piera Bontempo, Gemma D’Elia, Elisabetta Prada, Raffaele Vitale, Angelina Grammegna, Pierpaola Tannorella, Angela Sparago, Laura Pignata, Andrea Riccio, Silvia Russo, Giovanni Battista Ferrero, Alessandro Mussa

**Affiliations:** 1Department of Medical Sciences, University of Torino, 10126 Torino, Italy; 2Department of Public Health and Pediatric Sciences, University of Torino, 10126 Torino, Italy; 3Fondazione IRCCS Ca’ Granda Ospedale Maggiore Policlinico, 20122 Milan, Italy; 4Neonatology Unit, St. Orsola-Malpighi Polyclinic, Department of Medical and Surgical Sciences (DIMEC), University of Bologna, 40138 Bologna, Italy; 5Center for Rare Diseases and Birth Defects, Department of Woman and Child Health and Public Health, Fondazione Policlinico Universitario A. Gemelli, IRCCS, 00168 Rome, Italy; 6Rare Diseases and Medical Genetics Unit, University-Hospital Pediatric Department (DPUO), Bambino Gesù Children’s Hospital, IRCSS, 00165 Rome, Italy; 7Department of Pediatrics, Medical Faculty, “Sapienza” University of Rome, 00185 Rome, Italy; 8Department of Medicine, Surgery and Dentistry “Scuola Medica Salernitana”, Pediatrics Section, University of Salerno, 84081 Baronissi, Italy; 9Laboratory of Medical Genetics, Molecular Genetics Unit, Bambino Gesù Children Hospital, IRCCS, 00165 Rome, Italy; 10DAI Materno-Infantile, Università degli Studi di Napoli Federico II, 80138 Naples, Italy; 11Research Laboratory of Medical Cytogenetics and Molecular Genetics, IRCCS Istituto Auxologico Italiano, 20021 Milan, Italy; 12Institute of Genetics and Biophysics A. Buzzati-Traverso, Consiglio Nazionale delle Ricerche, 80131 Naples, Italy; 13Department of Environmental Biological and Pharmaceutical Sciences and Technologies (DiSTABiF), Università degli Studi della Campania “Luigi Vanvitelli”, 81100 Caserta, Italy; 14Department of Clinical and Biological Sciences, University of Torino, 10126 Torino, Italy; 15Pediatric Clinical Genetics, Regina Margherita Children Hospital, 10126 Torino, Italy

**Keywords:** Beckwith–Wiedemann syndrome spectrum, tumor, genomic imprinting, score

## Abstract

**Simple Summary:**

Beckwith-Wiedemann syndrome (BWSp) has recently been renamed to spectrum to reflect its diverse presentation and clinical features. In 2018, an international consensus developed a diagnostic approach and redefined clinical criteria, establishing a score above which a diagnosis can be made in case of a negative genetic test. We described a cohort of 831 patients to validate the efficacy of the 2018 consensus score for BWSp diagnosis, and to gather data on the performance of previous and current scoring systems, as well as the relationship between BWSp features, molecular tests, and the risk of cancer development.

**Abstract:**

Different scoring systems for the clinical diagnosis of the Beckwith–Wiedemann spectrum (BWSp) have been developed over time, the most recent being the international consensus score. Here we try to validate and provide data on the performance metrics of these scoring systems of the 2018 international consensus and the previous ones, relating them to BWSp features, molecular tests, and the probability of cancer development in a cohort of 831 patients. The consensus scoring system had the best performance (sensitivity 0.85 and specificity 0.43). In our cohort, the diagnostic yield of tests on blood-extracted DNA was low in patients with a low consensus score (~20% with a score = 2), and the score did not correlate with cancer development. We observed hepatoblastoma (HB) in 4.3% of patients with UPD(11)pat and Wilms tumor in 1.9% of patients with isolated lateralized overgrowth (ILO). We validated the efficacy of the currently used consensus score for BWSp clinical diagnosis. Based on our observation, a first-tier analysis of tissue-extracted DNA in patients with <4 points may be considered. We discourage the use of the consensus score value as an indicator of the probability of cancer development. Moreover, we suggest considering cancer screening for negative patients with ILO (risk ~2%) and HB screening for patients with UPD(11)pat (risk ~4%).

## 1. Introduction

Beckwith–Wiedemann Spectrum (BWSp, OMIM 130650) is the most common overgrowth disorder, affecting 1 in 10,340 people [1], and it is a model of human imprinting diseases and cancer predisposition. The most common features include neonatal hypoglycemia, macrosomia, macroglossia, lateralized overgrowth, omphalocele, predisposition to embryonal tumors (mostly Wilms tumor and hepatoblastoma), visceromegaly, renal abnormalities, and ear creases/pits [2,3,4,5,6]. The molecular bases of BWSp consist of epigenetic or genetic defects at two imprinting centers on chromosome 11p15.5: H19/IGF2:intergenic (IG) differentially methylated region (DMR) (also known as imprinting center 1, IC1) and KCNQ1OT1:transcriptional start site (TSS) DMR (also known as IC2). The result is an altered expression of the neighboring genes insulin-like growth factor 2 (*IGF2*), the long noncoding RNA H19, and cyclin-dependent kinase inhibitor 1C (*CDKN1C*) controlling fetal and postnatal growth and cell proliferation [7]. Four more common molecular defects are related to BWSp: loss of methylation at IC2 (IC2-LoM), occurring in 50–60% of cases; mosaic paternal uniparental isodisomy for part/all of chromosome 11 (UPD(11)pat) in 20–25%; gain of methylation at IC1 (IC1-GoM), in 5–10%; maternally inherited inactivating mutations of *CDKN1C* in 5–10%. Chromosomal rearrangements [8] of this region account for <1% [7,9]. Approximately 20% of clinically diagnosed patients lack a definite (epi)genotype.

Beckwith–Wiedemann syndrome has recently been renamed to spectrum in an international consensus [10] to emphasize its heterogeneity in a clinical presentation that variably includes several features in various degrees of severity. The spectrum includes so-called ‘classic’ and ‘atypical’ forms and spans to isolated lateralized overgrowth (ILO) [11]. Due to mild forms, it is well known that BWSp is underestimated from an epidemiological point of view. BWSp diagnosis is hampered by the low specificity of some features, very common in the general population, by difficulties in recognizing dysmorphisms in different populations [12], and by the level of tissue mosaicism of the underlying molecular defect, sometimes low or confined to tissues other than blood and difficult to be explored [13]. Indeed, some patients have negative molecular tests despite a clear-cut phenotype. Nevertheless, a prompt suspect and early diagnosis is the key to initiating specific follow-up and cancer screening, given that the majority of tumors in BWSp occur in early childhood [4]. When trying to standardize a diagnostic approach heterogeneous in the past, several scoring systems based on clinical criteria have been devised [2,3,14,15,16,17] and accumulated over time (Table 1), as well as management recommendations [18]. Lastly, in 2018, an international consensus [10] elaborated a diagnostic approach and redefined the clinical diagnostic criteria, separating for the first time the criteria for requesting molecular tests from those permitting a clinical diagnosis in the case of negative testing [10]. Based on their specificity, the consensus scoring system identified cardinal and supportive features, contributing 2 and 1 point, respectively. At least 2 points are required to trigger a specific molecular test and 4 to make a clinical diagnosis (notwithstanding a negative molecular test). Besides driving a genotype-based cancer screening, a positive molecular test allows making a diagnosis in cases with <4 points and supports the clinical diagnosis in those with ≥4 points. 

As pointed out in the international consensus [10], despite the extensive literature review, the scoring system criteria do not derive from a methodological or statistical review of case series but rather from a shared reasoned approach to the problem. Therefore, it is necessary to analyze the results deriving from implementing these recommendations in clinical practice to expand the evidence of the effectiveness and efficiency of such a system. This study aims at providing a systematic and statistical validation of the consensus criteria by evaluating its reliability in clinical practice, comparing it with the previously used criteria [2,3,14,15,16,17], analyzing its performance metrics against the outcomes of positive molecular tests and tumor development, as well as identifying the clinical features with higher diagnostic accuracy. 

## 2. Methods

Study cohort: This is a multicentric retrospective observational study that included patients from the main clinical genetics and rare disease centers in Italy, diagnosing and following patients with BWSp. Consent was obtained from the participants/parents, and this study was approved by the ethics committee (IRB 93/2021, Protocol 0070581-1 July 202). We asked the participant center to provide comprehensive data on the genotype and phenotype of the patients diagnosed with BWSp and filled out a spreadsheet including all the features described in BWSp. Efforts were made to collect medical records as completely as possible. Data were merged and entered into the database by assigning patients a unique identification number. 

The BWSp clinical score was calculated according to the international consensus criteria [10]. Scores and clinical diagnosis according to a previous version of diagnostic criteria, were also calculated according to each of the authors providing such definition [2,3,14,15,16,17] (Table 1). This is because, before the introduction of the consensus criteria there was heterogeneity in submitting patients to molecular tests and diagnosing BWSp clinically. We included in the analysis of this study cases diagnosed previous and after 2018, notwithstanding complete fulfillment of a clinical diagnosis of BWSp according to the consensus criteria. 

Genotyping: All the patients were tested on peripheral blood-extracted DNA and underwent analysis of the methylation level at the IC1 and IC2 by MS-MLPA [19,20], except for 20, who were tested by pyrosequencing [21]. Molecular testing on tissue other than blood was not carried out systematically by the various centers, so the outcome of these assessments has limited value for the purpose of this study. Overall, 14 patients with negative blood-extracted DNA tests were tested on DNA extracted from a skin biopsy of a hypertrophic body region (*n* = 13) or peritumoral tissue (*n* = 1).

MS-MLPA allows simultaneously detecting both hypermethylation at H19/IGF2:IG-DMR (IC1)-hypomethylation at KCNQ1OT1:TSS-DMR (IC2) and the copy number variants in these regions. In the case of both IC2-LoM and IC1-GoM with a proper copy number, a UPD(11)pat genotype was disclosed. Either microsatellite segregation or SNP array was then investigated to refine the extension of the disomy because when disomy involves the whole 11 chromosomes, paternal genome-wide UPD may occur [22]. We decided to rule out cases with GWpUPD/entire chromosome disomy a priori because (a) this study was multicenter, and this test was not performed consistently across the various referral centers that provided the data, and (b) these patients usually represent a very small subset of cases with a very different phenotype. Hence, we figured the criteria would not perform as well in such cases. In cases that are negative for MS-MLPA and with characteristics such as cleft palate, hereditary familiarity forms, diagnostic score > 8, or ≥4 with omphalocele, the screening of pathogenic variants in *CDKN1C* sequencing [22] completes the diagnostic flowchart. 

Statistical analysis: The differences between the clinical characteristics of the patients in the various molecular subgroups were evaluated with Fisher’s exact test or chi-square (for categorical variables) or Student’s *t*-test (after verification of homoscedasticity by Shapiro–Wilk test) for continuous variables. For the comparison of several groups, a one-way ANOVA test with a post hoc Bonferroni test was performed for the continuous variables. Correlations were tested with linear logistic regression (Pearson method). 

Positive and negative predictive value, sensitivity, specificity, and diagnostic accuracy for a positive methylation test were assessed for each of the clinical diagnostic scores by standard formulas for each of the items of the scoring systems proposed over time, used as a gold standard for the positive molecular test. The performance of each scoring system was also analyzed to identify a positive molecular test by a receiver operator characteristics (ROC) analysis, evaluated based on the area under the ROC curve (AUC).

A statistical significance threshold of two-sided *p* < 0.05 was used for all the tests. Data were analyzed using GraphPad Prism 8.0 packages (Graphpad Holdings, LLC, San Diego, CA, USA).

## 3. Results

**Genotype and phenotype**: We analyzed characteristics and results of the molecular tests of 831 patients with features within the BWSp submitted to specific molecular tests. Six hundred ninety-nine received a diagnosis of BWSp according to the consensus definition (524 with a positive test, 175 negatives with clinical criteria). One hundred thirty-two patients (15.9%) were negative for testing and had <4 points. The cohort distribution is summarized in Figure 1. In the inner circle, the cohort is divided into patients with positive and negative tests. However, the outer circle distinguishes patients with a clinical diagnosis and/or positive test from patients with less than 4 points and a negative test. Among the 831 patients, 322 had an IC2-LoM (38.7%, 61.0% of those with a positive molecular test), 52 had an IC1-GoM (6.3%, 9.9% with a positive molecular test), 138 had a UPD(11)pat (16.6%, 26.3% with a positive molecular test), 12 had a pathogenic variant in *CDKN1C* (1.4%, 2.3% with a positive molecular test), and 307 were negative (36.9%). In 519 patients, the molecular defect was found on blood-extracted DNA. In contrast, among the 14 patients tested on tissue-extracted DNA (all with score <4), 5 (35.7%) had positive tests (3 UPD(11)pat and 2 IC1-GoM). 

Table 2 summarizes the clinical features of the patients divided by molecular subtypes. Several differences in the features of the molecular subgroups, already reported elsewhere [4,23,24,25,26,27,28,29,30], were found: macroglossia was less represented in the UPD(11)pat group; facial naevus flammeus, ear anomalies, postnatal overgrowth, preterm birth, and abdominal wall defects (especially the major ones) were more common in the IC2-LoM and *CDKN1C* variant groups; lateralized overgrowth was very common in the UPD(11)pat group and rare in the *CDKN1C* variant one; fetal overgrowth, organomegaly, and polyhydramnios were more represented among patients with IC1-GoM; placentomegaly, cryptorchidism, and cleft palate were more common in the *CDKN1C* variant one; transient hypoglycemia was less common among negative patients; assisted reproduction and twinning were much more frequent in cases with IC2-LoM. As for concerns with tumors, the already assessed differences [5,31,32,33,34] were confirmed, with patients with IC1-GoM with a high risk (especially for Wilms Tumor (WT)), patients with UPD(11)pat one with an intermediate one (mostly WT and hepatoblastoma (HB)), patients with *CDKN1C* variants with a low risk (especially neuroblastic tumors), and patients with IC2-LoM with a very low one (for several different tumor types). 

Score: The patients had an average consensus score of 5.1 ± 2.1 (median 5.0) (Table 2). Patients negative for molecular tests had a score lower than those with a positive test (*p* < 0.001). However, this difference was rather due to a lower number of cardinal features (*p* < 0.001) than supportive ones. No differences in the score were found between the molecular subgroups but patients with IC1-GoM, despite a score similar to that of the other molecular subtypes, had fewer cardinal features (*p* = 0.043) and more supportive ones (*p* = 0.013) than the others. 

Figure 2 reports the distribution of patients’ clinical scores for each of the molecular subtypes, while, contrariwise, Figure 3 displays the molecular subgroups per each point of the consensus score. While we did not observe any differences in the score distribution in the four main molecular subgroups of BWSp, the distribution in cases with negative molecular tests was clearly skewed towards low scores. The probability of having a negative test diminished proportionately to a higher score (*p* < 0.001), ranging from 70% (score of 2 points) to 8% (score ≥ 10 points). Patients with a score < 4 were 210: 73 had a positive molecular test on blood-extracted DNA (34.8%), 20 with 2 points (23.8%), 53 with 3 points (40.7%), and 5 on tissue extracted-DNA (14 tested, 35.7%). One hundred thirty-two patients were negative for molecular tests and had diagnostic scores < 4. According to the consensus criteria, they cannot be diagnosed with BWSp. However, we included such cases in the cohort to provide a follow-up and comparison. A total of 56 had 2 points (8 with macroglossia, 48 with isolated lateralized overgrowth), 76 had 3 points deriving by a combination of one minor criterion with macroglossia (*n* = 11), lateralized overgrowth (*n* = 56), or omphalocele (*n* = 2), or from a combination of 3 minor criteria. 

**Tumors**: Table 3 details the 48 patients who developed a total of 50 tumors (2 patients had 2 tumors). A consistent difference in tumor risk was found in the four molecular defects, with the risk ranging from the highest (19.2%) in IC1-GoM to the lowest in IC2-LoM (1.6%) (*p* < 0.001). Tumor types were 26 WT (4 bilateral), 7 HB, 5 adrenal carcinomas (AK), 4 neuroblastomas (NB), 2 pancreatoblastomas (PB), 1 Sertoli cell tumor (SCT), 1 hepatocarcinoma (HC), 1 acute lymphoblastic leukemia (LLA), 1 Hodgkin lymphoma (LH), 1 rhabdomyosarcoma (RMS), and 1 pheochromocytoma (PCC). 

The score of patients without tumors was similar to the score of those with a tumor (not including points related to tumors, 5.4 ± 2.4 vs. 5.0 ± 2.0, *p* not significant). There was no correlation between the consensus score and the likelihood of developing a tumor (*r^2^* = 0.301, *p* = 0.124) nor differences between patients with more or less than 4 points (*p* = 0.160). Among the 132 negative patients with less than 4 points, 2 developed WT (1.5%, 1.9% of the patients with ILO), having 3 points and lateralized overgrowth as a cardinal feature.

Metrics of the scoring systems: Table 4 summarizes the performance metrics and characteristics of the several scoring systems for BWSp proposed over time. Each of the scoring systems based on clinical criteria was evaluated by ROC curves against the outcome of the molecular test and its ability to detect cases of cancer. For each scoring system, patients were divided into those with positive/negative molecular tests and with/without tumor development to assess the criteria’s ability to identify cases of interest. The sensitivity of the criteria ranged from 0.4 [14] to 0.88 [17] and specificity from 0.28 [17] to 0.86 [14]. The criteria from Ibrahim 2014 and the Consensus were those with the highest area under curve (AUC) and diagnostic accuracy in detecting patients with a positive molecular test. These data are presented visually in Figure 4, where the ROC curve for each scoring system is compared to the ROC curve for the consensus criteria (indicated with a red line). The performance against cancer development was tested, excluding from the score points deriving from the development of embryonal tumors. The consensus criteria allowed the diagnosis of BWSp in 40 of the 48 cases with cancer (83.3%), including 7 with negative molecular tests: the 8 cases not included were negative to molecular tests and scored less than 4 points. It is notable that LO remained a consistent clinical feature in patients with negative molecular analysis who developed tumors and had not previously reached the minimum score threshold for a clinical diagnosis of BWS, despite the possibility of mosaicisms. It is worth mentioning that none of these patients have yet been tested in alternative tissue, so we were unable to exclude mosaicisms.

Assisted Reproduction Technologies (ART): Eighty-eight patients (10.6%) were born through ART. Although there was no difference between the score of patients conceived through ART (5.5 ± 1.9) and those conceived naturally (5.1 ± 2.1), the former had, on average, a lower number of cardinal features than the latter (1.5 ± 0.6 vs. 1.7 ± 0.8, *p* = 0.050), and a higher number of supportive features (2.5 ± 1.4 vs. 2.2 ± 1.4, *p* = 0.036).

## 4. Discussion

The first international consensus group for diagnosing and managing BWSp was established in 2017. It provided the first standardized clinical criteria scoring system and testing algorithm to enhance the diagnosis and management of patients with BWSp. It has been observed that a major limitation of these recommendations was that they were derived from historical data, hampering the generalization to the whole BWSp population, as mostly inferred from ‘classic BWS’ cohorts [25]. In this study, we tried to apply to a large cohort of patients with BWSp or features within the BWSp, diagnostic criteria to provide evidence-based validation in the real-world context of such criteria. The population we analyzed included more than 800 subjects diagnosed with one of the entities of the BWSp (i.e., classic, atypical, (isolated)-lateralized overgrowth and with a variable association of cardinal features of BWS).

Our study group had an average score of 5.1, significantly lower than that of Duffy KA et al., another big cohort described with similar purposes (score 6.7) [25]. This is in part due to the fact that we included more patients with atypical and ILO phenotypes and also cases with some of the features of the BWSp without a formal diagnosis. Our group has a high percentage of cases negative to molecular tests (36.9%), as more frequently seen in patients with few clinical features of BWSp. Among the patients with positive molecular tests, the four molecular subgroups were roughly represented as expected from the literature, with a prevalence of the IC2-LoM (50–60%), ~25% of UPD(11)pat, 10% IC1-GoM, and <5% patients with *CDKN1C* mutations [7]. Differently from other studies [25], we found no differences between the molecular subgroups in the average score. However, patients with IC1-GoM tended to have fewer cardinal features and more supportive features than the others. In our cohort, also the genotype–phenotype correlations already reported [2,4,5,6,15,17,23,24,25,26,27,28,29,30,33] were observed in our patients.

In the cohort, the probability of having a negative molecular test result diminished proportionally to a higher score ranging from 70% in patients with 2 points to 8% in patients with more than 10 points. This difference can result from more localized tissue mosaicism in patients with less than 4 points. 

The likelihood of a positive molecular test on blood-extracted DNA was 50% for patients with a score of 4 points, 40% for patients with a score of 3 points, and less than 30% for patients with a score of 2 points. Among patients with a score of fewer than 4 points, the diagnostic yield of molecular tests on blood-extracted DNA was 34.8% (23.8% for patients with only 2 points). It would have been advisable to perform additional molecular investigations on these patients. As indicated in the consensus, molecular testing should be prioritized based on the cause that is most likely to be present. For example, a less severe phenotype with lateralized overgrowth may suggest mosaicism. In these cases, analyzing DNA from sources such as buccal swabs, cultures of fibroblasts, or cells of mesenchymal origin (obtained through surgical resection or excision of hyperplastic tissues) can help improve the detection rate for mosaic defects [10]. However, our study had relatively few cases in which tissue-extracted DNA was tested (14 out of 319 with negative molecular tests). These cases were largely limited to patients with molecular tests performed on blood-extracted DNA. This is likely due to the multicentric and retrospective nature of the study, as tissue-DNA testing was only recently introduced in clinical practice and was performed heterogeneously among the various centers. This might explain the slightly higher fraction of patients with IC2-LoM in our cohort, as most patients who are negative on blood testing but positive on tissue-extracted DNA usually have UPD(11)pat or IC1-GoM. It is interesting to note that more than one-third of the patients (5 out of 14) tested on tissue-extracted DNA were positive (a diagnostic yield of 35.7%). As previously reported, the molecular defects in these cases were frequently UPD(11)pat and IC1-GoM. Therefore, a first-tier approach with tissue testing would be at least comparable, if not even better, in this setting. Given the considerable share of negative cases below 4 points, a molecular approach from tissue-extracted DNA instead of blood could be advisable in patients with the possibility to define a regional involvement (e.g., tumor, ILO, pancreatic hyperplasia…), a testing approach which is commonly used in other conditions characterized by overgrowth and localized mosaicism [35]. In fact, an improvement in diagnostic performance has been documented by analyzing DNA from overgrowth tissue in these conditions, and the greater level of invasiveness in this situation would not only be justified by allowing a differential diagnosis towards other forms of overgrowth (e.g., PIK3CA-related overgrowth spectrum, vascular phenotype overlapping PIK3CA-related overgrowth spectrum with mutations in other genes) or body asymmetry (e.g., Silver–Russell syndrome) [35,36,37], but also by the opportunity to apply more precisely a targeted cancer screening [31] or management [38,39] based on the molecular lesion found within the BWSp. Further studies are needed to test this hypothesis and assess the best approach in such a condition, as well as the increase in the diagnostic yield by this approach.

With a view to defining the performance of the currently used diagnostic criteria for BWSp, we calculated the sensitivity and diagnostic accuracy of the consensus scoring system in detecting molecularly positive cases and cases with tumors. We also compared its performance using ROC curves with previous criteria suggested in the medical literature widely and heterogeneously employed in the years before 2018. The oldest criteria [14] were less sensitive and the most specific, allowing mostly diagnosing ‘Classic BWSp’. According to these criteria, nearly 60% of molecularly positive patients would not have been tested molecularly at all. The criteria by Gaston et al., 2001 [17] were more sensitive but were also less specific. Over time, the criteria published demonstrated an improvement in sensitivity at the expense of specificity. This was likely due to the fact that many of the BWSp features are not specific and very common in healthy children, as well as in the BWSp population. Adopting excessively broad and generous criteria would imply that patients with multiple low specificity features frequently in the population (e.g., fetal macrosomia, diastasis recti, or facial naevus simplex) would be diagnosed without being affected. There was a progressive improvement in the performance metrics of the criteria used across the years from 1994 to 2018: the positive predictive value was overall maintained, with a significant increase in the negative predictive value, resulting, therefore, in an increase in diagnostic accuracy over these 34 years. The more recent criteria from Ibrahim 2014 and the consensus were those with the highest AUC and diagnostic accuracy in detecting patients with a positive molecular test. The consensus criteria demonstrated superior performance in predicting cancer development, correctly identifying BWSp in 40 out of the 48 cases with cancer development (83.3%). This included seven cases with negative molecular tests and a clinical diagnosis. The eight cases not diagnosed by the consensus criteria were negative for molecular tests and scored less than 4 points, but all of these patients had ILO. Therefore, it may be advisable to consider molecular testing in other tissues in cases where a negative result is obtained. However, there is a strong suspicion of mosaicism. Additionally, according to consensus recommendations, alternative diagnoses should be considered if all molecular tests are negative.

Indeed, the main novelty introduced by these criteria was to establish a different score threshold to trigger the molecular analysis (≥2 points) and to formalize a clinical diagnosis of BWSp even with a negative molecular test (≥4 points). This expedient allowed obviating the intrinsic nonspecificity of some of the clinical features of BWSp, which are in themselves very common in the general population and, therefore, currently used as support criteria. In the future, similar reasoning could lead to implementation among the supporting criteria also the use of ART, which is significantly more common in subjects with BWSp [40] although frequent in the population. Future studies might clarify if this will result in a further improvement in the score performance metrics. 

Regarding ART, IC2-LoM is the subgroup with the highest rate of ART (16.1%). However, it is worth noting that a significant proportion of patients in the other subgroups were also born through ART. These data support the previously mentioned hypothesis of a connection between ART and BWS, as highlighted by Brioude et al. [10]. Moreover, it has been previously pointed out that many patients conceived via ART were characterized in the atypical or ILO groups [25] or by a less severe presentation [41]. Here we confirmed that, although they have an average score similar to that of patients naturally conceived, ART-conceived patients have less commonly cardinal and more supportive features. This further confirms that such patients less frequently belong to the “classic” BWSp group but rather are more frequently “atypical” [41]. This observation further corroborates the employment of ART usage as a supportive criterion to be implemented into the score in the future [42]. 

The most relevant concern for patients with BWSp is cancer development: managing cancer risk in this population requires specific screening programs for the early detection of tumors to reduce the treatment burden and improve outcomes [43,44]. These cancer screening programs can be effectively implemented in cases diagnosed with BWSp and demonstrate the best balance between benefit and medicalization when genotype-based [31]. However, a consistent portion of patients is diagnosed only after a tumor’s development, which diminishes the benefits of tumor screening. Some have no or very mild phenotypes [34,45]. With this view, the currently employed consensus score represents a great improvement, as it proves much more effective than the previous criteria in recognizing patients who will develop cancer. The consensus score allows diagnosing with BWSp and screening for tumor development in 83% of the cases that will develop cancer later. The main objective of the diagnostic score is to allow the carrying out of an empirical classification of the patient aimed at the follow-up and mainly at the adoption of a correct screening strategy. The score did not miss any case with a positive molecular test, proving itself sensitive enough to allow diagnosing all the positive patients. Interestingly, we observed no tendency for patients who develop tumors to have a high score. The score did not predict the probability of developing tumors, so it should not be used for this purpose or for stratifying patients based on their cancer risk. Indeed, many cases with tumors (17/48) occurred in patients with <4 points.

Among the 132 patients with less than 4 score points and negative molecular tests we included in our cohort, most (104, 78.8%) had LO as the sole or cardinal feature, and 2 (all with LO) developed a WT (1.9%). This observation has relevant management implications. Based on different approaches to the “acceptable risk” in different healthcare systems, the consensus recommendation adopted a 5% tumor risk cutoff to advise tumor surveillance [10]. In comparison, the American Association for Cancer Research (AACR) maintained a more conservative approach using a 1% cutoff [46]. This resulted in different approaches in different countries, with most clinicians in the U.S.A. continuing to screen all patients with BWSp. However, most EU countries screen only patients at high risk (IC1-GoM and UPD(11)pat). Currently, there are no clear recommendations concerning cancer screening in cases with LO due to the few studies that focused on this clinical entity [26,45,47,48]. Based on our results, patients with negative molecular tests and LO with <4 points at the score, having a ~2% risk of developing cancer, are under the 5% screening threshold for healthcare systems adopting a high acceptable risk of tumor and above the 1% threshold suggested by the AACR. Concerning cancer screening, a final observation can be carried out: HB developed in 4.3% of patients with UPD(11)pat, a fraction significantly higher than that of the other molecular subgroups. This finding further supports our previous recommendation to screen at least patients with UPD(11)pat by alpha-fetoprotein [49,50,51]. It suggests that screening should be conducted until 30 months [52].

## 5. Conclusions

In conclusion, this study documented the very high performance and effectiveness of the currently employed diagnostic criteria for BWSp, supporting a widespread implementation. Moreover, our study proposes several further hints for the diagnosis and management of patients with BWsp: (a) it could be an effective strategy to begin molecular testing from tissue-extracted DNA in patients with <4 points and an identifiable affected body region, as the diagnostic yield of tests on blood-extracted DNA, is low, (b) adding ART among the supportive criteria of the scoring system might lead to an improvement in the score performance, (c) clinicians should refrain from using the score as an indicator of the probability of cancer development, (d) patients with ILO and <4 points with negative molecular test have a cancer risk between 1% and 5%, and screening in such condition should be applied in accordance with specifics of local healthcare systems, and (e) our data support the screening of HB in patients with UPD(11)pat.

## Figures and Tables

**Figure 1 cancers-15-00773-f001:**
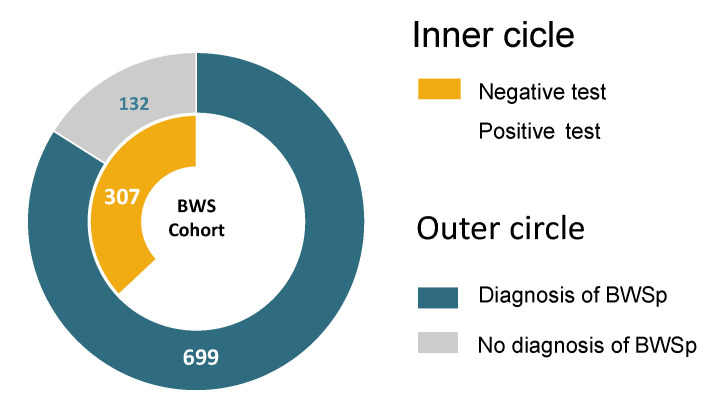
Distribution of the cohort. In the inner circle, patients with a negative test are in yellow, and patients with a positive test are blue; in the outer circle, patients with less than 4 points and a negative test are in gray; patients with a clinical diagnosis and/or positive test are in blue.

**Figure 2 cancers-15-00773-f002:**
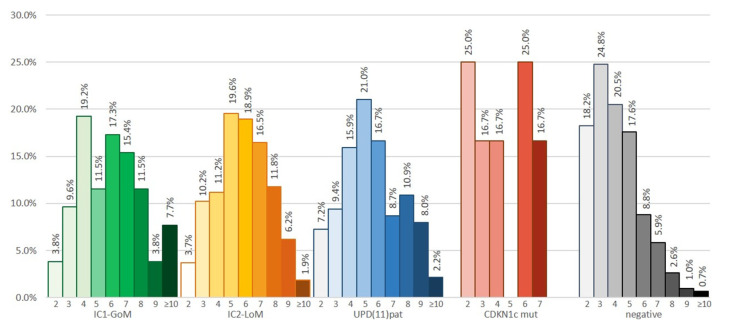
Molecular subgroups per each point of the consensus score [10]. Percentages of the negative cases (in gray) refer to the total. Percentages in the four subgroups refer to the total of positive cases, including gain of methylation at imprinting center 1 (IC1-GoM, in green), loss of methylation at imprinting center 2 (IC2-LoM in orange), paternal uniparental disomy of the 11p15.5 chromosomal region (UPD(11)pat, in blue), and pathogenic variants in *CDKN1C* (*CDKN1c* mut, in red).

**Figure 3 cancers-15-00773-f003:**
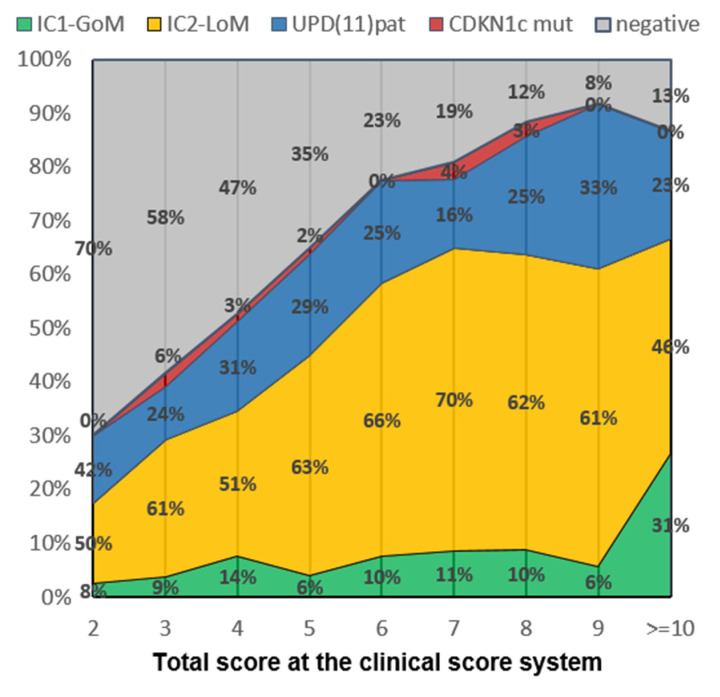
Molecular breakdown for each of the points of the consensus scoring system [10]. Percentages of the negative cases (in gray) refer to the total, while percentages in the four subgroups refer to the total of positive cases including gain of methylation at imprinting center 1 (IC1-GoM, in green), loss of methylation at imprinting center 2 (IC2-LoMm in orange), paternal uniparental disomy of the 11p15.5 chromosomal region (UPD(11)pat, in blue), and pathogenic variants in *CDKN1C* (CDKN1c mut, in red).

**Figure 4 cancers-15-00773-f004:**
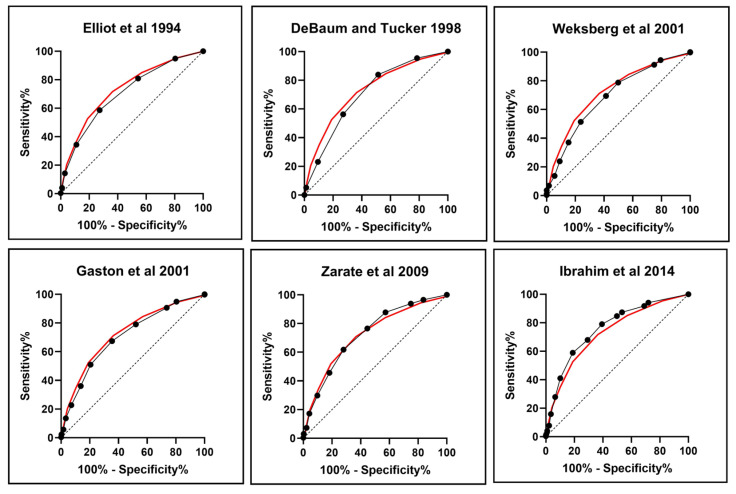
Receiver Operating Characteristic (ROC) curves of the several scoring systems for BWSp proposed over time, evaluated against the outcomes of the molecular tests [2,3,10,14,15,16,17]. Each of the ROC curves of the various scoring system (in black) is compared with the ROC curve of the consensus criteria by Brioude et al. 2018 [10] (red line). The ROC curve characteristics, area under the curve (AUC), standard error (st.Err), 95% Confidence Interval (95%CI), and *p*-value are given in Table 4.

**Table 1 cancers-15-00773-t001:** Beckwith–Wiedemann syndrome (BWS) and spectrum (BWSp) clinical diagnostic criteria proposed over time by several authors.

	Major Features	Minor Features	Clinical Diagnosis of Beckwith–Wiedemann Syndrome
Elliott et al., 1994 [14]	Abdominal wall defect Macroglossia Macrosomia ^a^	Ear creases/pits Facial naevus simplex Lateralized overgrowth Hypoglycemia Nephromegaly	At least three major features, or two major features plus three or more minor features
DeBaun and Tucker 1998 [15]	Abdominal wall defect Ear creases/pits Hypoglycemia Macroglossia Macrosomia		At least two major features
Weksberg et al., 2001 [3]	Abdominal wall defect Ear creases/pits Embryonal tumors Lateralized overgrowth Macroglossia Macrosomia	Hypoglycemia Organomegaly Renal malformation	At least three major features, or two major features and one or more minor features
Gaston et al., 2001 [17]	Abdominal wall defect Macroglossia Macrosomia Organomegaly	Ear creases/pits Facial naevus simplex Lateralized overgrowth Hypoglycemia	Complete and incomplete BWS syndrome classification. Complete—at least three major features. Incomplete—less than three major features and one or more minor features
Zarate et al., 2009 [16]	Abdominal wall defect Macroglossia Macrosomia	Cardiomegaly Ear creases/pits Facial naevus simplex Lateralized overgrowth Hypoglycemia Mid-face hypoplasia Polyhydramnios	At least three major features, or two major features and one or more minor features
Ibrahim et al., 2014 [2]	Macroglossia Omphalocele Organomegaly Macrosomia Facial naevus simplex Lateralized overgrowth Hypoglycemia		Macroglossia 2.5 pts Omphalocele 1.5 pts Organomegaly 1 pts Macrosomia 1 pts Facial naevus simplex 1 pts Lateralized overgrowth 0.5 pts Hypoglycemia 0.5 pts Clinical diagnosis if score is ≥3.5 pts
Brioude et al., 2018 [10]	Macroglossia Omphalocele Lateralized overgrowth Multifocal and/or bilateral Wilms tumor or nephroblastomatosis Hyperinsulinism ^b^ Pathology findings ^c^	Macrosomia Facial naevus simplex Polyhydramnios and/or placentomegaly Ear creases and/or pits Transient hypoglycemia ^d^ Typical BWSp tumors ^e^ Nephromegaly and/or hepatomegaly Umbilical hernia and/or diastasis recti	Clinical diagnosis if score is ≥4. Patients with a score of ≥2 merit genetic testing for investigation and diagnosis of BWS. Patients with a score of <2 do not meet the criteria for genetic testing.

^a^ Macrosomia: Birthweight >2 SDS; ^b^ Hyperinsulinism: prolonged hypoglycemia in the context of elevated insulin levels that last >1 week and/or require escalated treatment; ^c^ Pathology findings: adrenal cortex cytomegaly, placental mesenchymal dysplasia or pancreatic adenomatosis; ^d^ Transient hypoglycemia: hypoglycemia that lasts <1 week and resolves without the need for further intervention; ^e^ Typical BWSp tumors: neuroblastoma, rhabdomyosarcoma, unilateral Wilms tumor, hepatoblastoma, adrenocortical carcinoma or phaeochromocytoma.

**Table 2 cancers-15-00773-t002:** Clinical features and diagnostic criteria of the study group divided by molecular subtype.

	Total	IC1-GoM	IC2-LoM	UPD(11)pat	*CDKN1C*	Negative	*p* Value
Total	831	52 (6.3%)	322 (38.7%)	138 (16.6%)	12 (1.4%)	307 (36.9%)	
Females	445	27 (51.9%)	187 (58.1%)	67 (48.6%)	6 (50%)	158 (51.5%)	0.321
Males	386	25 (48.1%)	135 (41.9%)	71 (51.4%)	6 (50%)	149 (48.5%)	0.362
**Cardinal and suggestive features**							
Macroglossia	557	42 (80.8%)	276 (85.7%)	92 (66.7%)	10 (83.3%)	137 (44.6%)	<0.001
Abdominal wall defects	476	28 (53.8%)	214 (66.5%)	68 (49.3%)	10 (83.3%)	156 (50.8%)	<0.001
Umbilical hernia	151	6 (11.5%)	64 (19.9%)	29 (21%)	5 (41.7%)	47 (15.3%)	0.060
Diastasis recti	226	20 (38.5%)	80 (24.8%)	30 (21.7%)	0 (0%)	96 (31.3%)	0.01
Omphalocele	99	2 (3.8%)	70 (21.7%)	9 (6.5%)	5 (41.7%)	13 (4.2%)	<0.001
Lateralized overgrowth	551	29 (55.8%)	178 (55.3%)	119 (86.2%)	2 (16.7%)	223 (72.6%)	<0.001
Neonatal hyperinsulinism	11	1 (1.9%)	3 (0.9%)	5 (3.6%)	0 (0%)	2 (0.7%)	0.119
Fetal overgrowth	334	36 (69.2%)	142 (44.1%)	62 (44.9%)	6 (50%)	88 (28.7%)	<0.001
Facial naevus simplex	256	10 (19.2%)	139 (43.2%)	32 (23.2%)	6 (50%)	69 (22.5%)	<0.001
Polyhydramnios	100	11 (21.2%)	55 (17.1%)	17 (12.3%)	1 (8.3%)	16 (5.2%)	<0.001
Placentomegaly	7	0 (0%)	5 (1.6%)	1 (0.7%)	1 (8.3%)	0 (0%)	0.011
Ear anomalies	296	17 (32.7%)	142 (44.1%)	47 (34.1%)	6 (50%)	84 (27.4%)	<0.001
Transient hypoglycemia	226	18 (34.6%)	104 (32.3%)	52 (37.7%)	3 (25%)	49 (16%)	<0.001
Organomegaly	176	30 (57.7%)	75 (23.3%)	26 (18.8%)	1 (8.3%)	44 (14.3%)	<0.001
Typical tumors	44	9 (17.3%)	6 (1.9%)	17 (12.3%)	1 (8.3%)	11 (3.6%)	<0.001
Wilms tumor–multifocal	22–4	8 (15.4%)–1 (1.9%)	0 (0%)	6–0 (4.3%)	0 (0%)	8 (3.6%)–3 (1%)	<0.001
Hepatoblastoma	7	0 (0%)	1 (0.3%)	6 (4.3%)	0 (0%)	0 (0%)	<0.001
Neuroblastoma	4	0 (0%)	1 (0.3%)	2 (1.4%)	1 (8.3%)	0 (0%)	<0.001
Pancreatoblastoma	2	1 (1.9%)	0 (0%)	1 (0.7%)	0 (0%)	0 (0%)	0.061
Rhabdomyosarcoma	1	0 (0%)	1 (0.3%)	0 (0%)	0 (0%)	0 (0%)	0.812
Adrenal gland carcinoma	4	0 (0%)	2 (0.6%)	2 (1.4%)	0 (0%)	1 (0.3%)	0.656
Pheochromocytoma	1	0 (0%)	0 (0%)	0 (0%)	0 (0%)	1 (0.3%)	0.789
**Other features**							
Other tumors	3	0 (0%)	0 (0%)	1 (0.7%)	0 (0%)	2 (0.6%)	0.668
Cleft palate	11	0 (0%)	8 (2.5%)	0 (0%)	1 (8.3%)	2 (0.7%)	0.022
Assisted reproduction	88	3 (5.8%)	52 (16.1%)	14 (10.1%)	1 (8.3%)	18 (5.9%)	<0.001
Postnatal overgrowth	267	19 (36.5%)	130 (40.4%)	40 (29%)	7 (58.3%)	71 (23.1%)	<0.001
Cryptorchidism	47	5 (20%)	19 (14.1%)	5 (7%)	4 (66.7%)	11 (7.4%)	<0.001
Twin pregnancy	50	1 (1.9%)	22 (6.8%)	1 (0.7%)	0 (0%)	12 (3.9%)	0.032
Preterm delivery	212	9 (17.3%)	105 (32.6%)	21 (15.2%)	7 (58.3%)	70 (22.8%)	<0.001
Gestational age (weeks)	37.0 ± 2.9	37.3 ± 2.7	37.0 ± 2.6	37.9 ± 2.4	33.2 ± 4.1 *	36.4 ± 3.4	<0.001
Consensus score (mean ± SD)	5.1 ± 2.1	5.9 ± 2.2	5.8 ± 1.9	5.5 ± 2.1	5.3 ± 2.0	4.1 ± 1.7	<0.001
Cardinal features (mean ± SD)	1.5 ± 1.7	1.6 ± 0.8 ⁑	1.7 ± 0.7	1.8 ± 0.7	1.6 ± 0.7	1.3 ± 0.6	<0.001
Suggestive features (mean ± SD)	2.2 ± 0.7	3.1 ± 1.7 •	2.5 ± 1.4	2.3 ± 1.5	2.5 ± 1.6	2.5 ± 1.6	<0.001

Abbreviations: Gain of methylation at imprinting center 1 (IC1-GoM), loss of methylation at imprinting center 2 (IC2-LoM), paternal uniparental disomy of the 11p15.5 chromosomal region (UPD(11)pat), pathogenic variants in *CDKN1C* (*CDKN1C* mutation), standard deviation (SD). ⁑ *p* = 0.043; • *p* = 0.013; * *p* < 0.001, excluding patients with negative molecular tests.

**Table 3 cancers-15-00773-t003:** Patients with cancer and tumor type divided by molecular group and consensus score.

ConsensusScore ^a^	*n* (%)	IC1-GoM	IC2-LoM	UPD(11)pat	*CDKN1C* Mutation	Negative
Any scoresScores	48/831 (5.8%)	10 (19.2%)	5 (1.6%)	17 (12.3%)	1 (8.3%)	15 (4.9%)
2	5/85 (5.9%)	-	-	1 AK, 2 HB	-	2 WT
3	12/137 (8.8%)	1 WT	1 RMS	1 AK, 1 WT, 1 HB, 1 leu	-	2 WT + 1 bWT, 1 AK, 1 LH, 1 PCC
4	2/124 (1.6%)	-	-	1 WT, 1 NB	-	-
5	5/156 (3.2%)	-	1 AK	1 PB * 1 NB *	-	2 WT + 1 bWT
6	7/123 (5.7%)	2 WT + 1 bWT	1 AK ′, 1 SCT ′	-	1 NB	1 HCC, 1 WT
7	8/95 (8.4%)	1 PB 1 WT	1 HB, 1 NB	2 WT, 1 HB	-	1 WT
8	4/64 (6.2%)	1 WT	-	2 WT	-	1 bWT
9	2/35 (5.7%%)	1 WT	-	1 HB	-	-
≥10	3/12 (25%%)	2 WT	-	1 HB	-	-

^a^ The score in the table has been calculated without taking into account the points related to tumors; *,′ these tumors occurred in the same patient. Abbreviations: gain of methylation at imprinting center 1 (IC1-GoM), loss of methylation at imprinting center 2 (IC2-LoM), paternal uniparental disomy of the 11p15.5 chromosomal region (UPD(11)pat), pathogenic variants in *CDKN1C* (CDKN1C mutation), Wilms tumors (WT), bilateral Wilms tumors (bWT), hepatoblastomas (HB), adrenal carcinomas (AK), neuroblastomas (NB), pancreatoblastomas (PB), Sertoli cell tumor (SCT), hepatocarcinoma (HC), acute lymphoblastic leukemia (LLA), Hodgkin lymphoma (LH), 1 rhambdomyosarcoma (RMS), and 1 pheochromocytoma (PCC).

**Table 4 cancers-15-00773-t004:** Performance metrics and characteristics of the several scoring systems for BWSp proposed over time evaluated against the outcomes a) molecular test and b) cancer development: area under the curve (AUC), standard error (st.Err), 95% Confidence Interval (95%CI), *p*-value, sensitivity (Se), specificity (Sp), positive and negative predictive value (PPV, NPV), and accuracy (ACC).

	Elliott et al., 1994 [14]	DeBaun and Tucker 1998 [15]	Weksberg et al., 2001 [3]	Gaston et al., 2001 [17]	Zarate et al., 2009 [16]	Ibrahim et al., 2014 [2]	Brioude et al., 2018 [10]
**Performance against a Positive/Negative Molecular Test**							
Patients with/without clinical criteria	253/578	598/233	565/266	684/147	455/376	535/296	621/210
Positive test	210/314	440/84	412/112	462/62	344/180	414/110	446/78
IC1-GoM	27/25	43/9	44/8	45/7	38/14	42/10	45/7
IC2-LoM	136/186	282/40	258/64	290/32	222/100	268/54	277/45
UPD(11)pat	42/96	103/35	102/36	118/20	76/62	95/43	115/23
*CDKN1C* mutation	5/7	12/0	8/4	9/3	8/4	9/3	9/3
Negative test	43/264	158/149	153/154	222/85	111/196	121/186	175/132
**ROC Analysis**							
Area under curve (AUC)	0.706	0.706	0.692	0.704	0.724	0.761	0.731
st.Err	0.018	0.019	0.019	0.019	0.018	0.017	0.018
95% Confidence Interval	0.67–0.74	0.69–0.74	0.65–0.73	0.69–0.74	0.68–0.76	0.73–0.79	0.70–0.77
*p* value	<0.001	<0.001	<0.001	<0.001	<0.001	<0.001	<0.001
Sensitivity	0.40	0.84	0.79	0.88	0.66	0.79	0.85
Specificity	0.86	0.49	0.5	0.28	0.64	0.61	0.43
Positive Predictive Value (PPV)	0.83	0.74	0.73	0.68	0.76	0.77	0.72
Negative Predictive Value (NPV)	0.46	0.64	0.58	0.58	0.52	0.63	0.63
Diagnostic Accuracy (ACC)	0.57	0.71	0.68	0.66	0.65	0.72	0.69
**Performance against Tumor Development (*n* = 48) ***							
Patients with tumor diagnosed	16 (33.3%)	30 (62.5%)	18 (37.5%)	40 (83.3%)	24 (50.0%)	27 (56.8%)	31 + 9 ⁑ (83.3%)
With positive molecular tests	13	25	15	28	20	22	24 + 9 ⁑
With negative molecular tests	3	5	3	12	4	5	7
Patients with tumor missed	32 (66.7%)	18 (37.5%)	30 (62.5%)	8 (16.7%)	24 (50.0%)	21 (43.2%)	8 (16.7%)
With positive molecular tests	20	8	18	5	13	11	0
With negative molecular tests	12	10	12	3	11	10	8

* Scores were calculated excluding the points attributable to the development of tumor in the criteria including this data among the features for the diagnosis; ⁑ 31 patients were diagnosed with >4 points, 9 more patients with tumor would have been caught because positive to molecular testing with a score ≥ 2, 8 patients had negative molecular tests and score of 2–3; Abbreviations: gain of methylation at imprinting center 1 (IC1-GoM), loss of methylation at imprinting center 2 (IC2-LoM), paternal uniparental disomy of the 11p15.5 chromosomal region (UPD(11)pat), pathogenic variants in *CDKN1C* (CDKN1C mutation), and standard error (st.Err).

## Data Availability

The data that support the findings of this study are available from the corresponding author, upon reasonable request.

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
