# Peer review of "Performance Metrics of the Scoring System for the Diagnosis of the Beckwith–Wiedemann Spectrum (BWSp) and Its Correlation with Cancer Development"

_cancers, 2023, doi:10.3390/cancers15030773_

Round 1
Reviewer 1 Report
This paper is a very well written multi-centre validation of the BWS consensus guidelines, based on a lage number of cases
The results are significant for everyone using these guidelines in clinical care of BWS patients
Minor remarks are listed in the word file attached.

Reviewer 2 Report
Dr Luca et al reports a clinical evaluation of a new scoring system for the diagnosis of Beckwith Wiedemann spectrum that has been defined in 2017. The results have been obtained from a large clinically and biologically well defined cohort of patients. This paper brings important information about the capacity of this scoring system to detect patients with a molecular anomaly, and the patients with a high risk of cancer development (justifying cancer screening). The paper is well written, with up-to-date references.
However I have some concerns:
1) representing the results as table make things somehow difficult to understand:
- it is not clear how the patients are distributed between the different "conditions" (positive/negative for a molecular test, positive/negative for the clinical scoring system). A graphical representation of the cohort could be helpful.
- for table 2: symptoms might be sorted as cardinal/suggestive within the table according to the clinical scoring system. Features that are outside from the scoring system might be presented in an other table.
- for table 4: a graphical representation of the ROC curves might be interesting to have a visual representation of Specificity and Sensitivity of each scoring system.
2) I have some questions about some of the categories.
- the main text mentions 8 patients with a tumour but a clinical score less than 4 and a negative molecular test (thus not defined as BWSp according to the scoring system presented here). Regarding table 3, 9 tumours are mentioned for negative patients (which seems discrepant from the 8 patients mentioned in the main text). I am wandering how many of these 8 or 9 patients have isolated lateralized overgrowth (ILO)? Have some of these patients been tested in an alternative tissue?
- The authors still present the patients with a score less than 4 and a negative molecular test as BWSp, as they have been refered for a clinical suspicion of BWS. However, a strong clinical overlap has been demonstrated between various overgrowth conditions, including generalized (Simpson Golabi Behmel, Sotos, Weaver syndromes...) or segmental overgrowth (PIK3CA related overgrowth syndromes), some of them being associated with tumours that also overlap with the tumours associated with BWSp. Thus, the 8 or 9 patients with tumours that have been "missed" with the scoring system might have been "misdiagnosed". I would like the authors to discuss about the possibility for those patients to have an alternative diagnosis, as the methods part does not mention any molecular analysis for alternative diagnosis as suggested in the consensus paper from Brioude et al in 2018. I would also like the authors to give more clinical details about the patients who developed a tumour but with no molecular anomaly (at least for the 8/9 patients with a score of 2 or 3).
- The prevalence of tumours presented in this cohort is slightly lower than those that have been reported elsewhere. I would like the authors to mention the age at molecular diagnosis and the age at last clinical evaluation (if available), as some patients might be of very young age and thus still being in the high risk period. If not easily available, I would like the authors to discuss this point.
3) Regarding the discussion part, I do have concerns about the % of positive results in leucocytes vs tissues. The way the authors present this result (34.8% vs 35.7%) make think that the two procedures have a same rate of positivity. My concern is that patients with a molecular anomaly detected in blood did not probably have a test in tissues, but most of them will probably carry the anomaly in both leucocytes and tissues. I would appreciate the authors to revise this part of the discussion
4) Regarding ART (which is somehow out from the main objective of this paper), I would appreciate the authors to comment about the % of ART in the different subgroups. The high rate of ART in the subgroup of patients with IC2 LoM is obvious (16.1%, table 2). However, it is less obvious that the rate of ART in the other subgroups is increased compared to the rate of ART in the general population.
5) Typos and minor modifications:
Line 70 : please remove the coma between "at" and "IC2".
Line 158 : please write "specificity" instead of "specific"
Line 186 : please write "hypoglycemia" instead of "hypoglicemia"
Line 244 : please write "rhabdomyoracoma" instead of "rhambdomyosarcoma"
Numbers on the x-axis of figure 1 are not aligned with the histograms.
